# Nuclear Magnetic Resonance Metabolomics Biomarkers for Identifying High Risk Patients with Extranodal Extension in Oral Squamous Cell Carcinoma

**DOI:** 10.3390/jcm9040951

**Published:** 2020-03-30

**Authors:** Cheng-Kun Tsai, Chien-Yu Lin, Chung-Jan Kang, Chun-Ta Liao, Wan-Ling Wang, Meng-Han Chiang, Tzu-Chen Yen, Gigin Lin

**Affiliations:** 1Clinical Metabolomics Core Lab, Chang Gung Memorial Hospital at Linkou, Taoyuan 333, Taiwan; klem.tsaick@gmail.com (C.-K.T.); 0914.neo@gmail.com (M.-H.C.); 2Imaging Core Lab, Institute for Radiological Research, Chang Gung Memorial Hospital at Linkou, Taoyuan 333, Taiwan; 3Department of Radiation Oncology, Chang Gung Memorial Hospital at Linkou, Taoyuan 333, Taiwan; qqvirus1022@gmail.com; 4Head and Neck Oncology Team, Chang Gung Memorial Hospital at Linkou, Taoyuan 333, Taiwan; handneck@gmail.com (C.-J.K.); liaoct@adm.cgmh.org.tw (C.-T.L.); 5Particle Physics and Beam Delivery Core Lab, Institute for Radiological Research, Chang Gung Memorial Hospital at Linkou, Taoyuan 333, Taiwan; 6Department o Otorhinolaryngology, Head and Neck Surgery, Chang Gung Memorial Hospital at Linkou, Taoyuan 333, Taiwan; 7Department of Nuclear Medicine, Chang Gung Memorial Hospital at Linkou, Taoyuan 333, Taiwan; sweet216014@hotmail.com; 8Department of Medical Imaging and Intervention, Chang Gung Memorial Hospital at Linkou, Taoyuan 333, Taiwan

**Keywords:** oral squamous cell carcinoma, extranodal extension, metabolomics, nuclear magnetic resonance, metabolite, biomarker, metabolic pathway

## Abstract

Extranodal extension (ENE) is an independent adverse prognostic factor in oral squamous cell carcinoma (OSCC), and is difficult to identify preoperatively. We aimed to discover biomarkers for high risk patients with ENE. Tandem tissue, plasma, and urine samples of 110 patients with OSCC were investigated through 600-MHz nuclear magnetic resonance (NMR) metabolomics analysis. We found that the levels of creatine, creatine phosphate, glycine, and tyramine in plasma significantly decreased in stage IV ENE positive OSCC compared with stage IV ENE negative OSCC. To understand the underlying mechanism behind the alteration of plasma metabolites, our tissue analysis revealed that the carnitine level significantly increased in tumors but significantly decreased in the adjacent normal tissue in advanced stage OSCC, in addition to decreased levels of alanine and pyruvate in tumor tissues. The global metabolomics analysis on tumor tissues also showed that stage IV tumors with an ENE positive status demonstrated higher levels of aspartate, butyrate, carnitine, glutamate, glutathione, glycine, glycolate, guanosine, and sucrose but lower levels of alanine, choline, glucose, isoleucine, lactate, leucine, myo-inositol, O-acetylcholine, oxypurinol, phenylalanine, pyruvate, succinate, tyrosine, valine, and xanthine than tumors with an ENE negative status. We concluded that metabolomics alterations in tumor tissues correspond to an increase in the tumor stage and are detectable in plasma samples. Metabolomic alterations of OSCC can serve as potential diagnostic markers and predictors of ENE in patients with stage IV OSCC.

## 1. Introduction

More than 300,000 patients are annually estimated to have oral squamous cell carcinoma (OSCC) worldwide, and the mortality rate is 48%. It is an aggressive cancer that can infiltrate and metastasize highly, with a reported five-year survival rate of approximately 50% [1,2]. However, despite new treatments, there has been no marked improvement in five-year survival rates over the past few decades [3]. The American Joint Committee on Cancer (AJCC) Cancer Staging Manual, eighth edition, marked a paradigm shift in the staging system for head and neck cancers by introducing several novel considerations in distinct therapeutic ramifications [4]. In OSCC, the nodal staging recommendations have upstaged extranodal extension (ENE) of metastatic nodes, which has been reported as a high-risk adverse feature associated with worse survival [5]. The presence of ENE is an independent adverse prognostic factor in OSCC [6,7], and it significantly decreases survival [8,9] as well as locoregional and distant control [10,11].

Cancer cells have been reported to have altered metabolism in glycolysis, which is called the “Warburg effect”, and several other metabolic pathways are yet to be elucidated. Accumulated evidence has shown that the host-environmental manipulations could modulate the metabolism in OSCC and potentially serve as clinic prognostic indicators or developing adjuvant anti-cancer therapy [12]. Metabolomics may be a comprehensive approach to clarify the entire metabolic system that plays a role in OSCC. Preliminary metabolomic analysis of OSCC tissue not only revealed differences between the metabolomic characteristics of cancer and normal tissues but also suggested that specific metabolic systems play a role in OSCC tissues [13]. Downregulated amino acid levels might be associated with enhanced energy metabolism, and the upregulation of the biosynthetic pathway might reflect the desired cellular proliferation in cancer cells [14]. Metabolites can be transferred through tissue compartments to the blood stream, promoting tumor growth and aggressive tumor behavior [15]. The plasma metabolomics profile of Sprague–Dawley rats demonstrated time-dependent changes during various stages of OSCC, which provides insight into the sequential mechanisms of OSCC development [16]. Metabolomic biomarkers represent the host–environment interaction and are currently of interest because they may constitute the most essential step of diagnosis. In the future, specific and personalized diagnoses should accompany treatment to increase the chances of curing the disease [17]. Detailed metabolomics analysis of tissue, plasma, and urine samples might enhance our understanding of the pathogenesis, thus enabling the development of new prevention and treatment strategies for OSCC [18].

The aim of the present study is to identify a surrogate marker for ENE prediction through the metabolomics investigation of tandem tissue, plasma, and urine samples of patients with OSCC.

## 2. Materials and Methods

### 2.1. Patients and Histopathology

An institutional review board approved the protocol of this prospective study (IRB103-6164C, IRB104-1143C, IRB104-2820C), and informed consent was obtained from patients with OSCC at a tertiary referral center, with a dedicated OSCC interdisciplinary team to conduct patient selection. From December 2014 to November 2017, we screened a consecutive cohort of 115 patients with OSCC, and excluded five patients with unspecified ENE status (stage I, n = 1; stage II, n = 2; stage III, n = 2). The final analysis was performed in 110 patients. We only included patients with oral cavity cancers in this study. Tumors from soft palate or tongue base origin were categorized into oropharyngeal cancer hence not included in this study. We did not enrolled patients with soft palate or tongue base origin which is categorized into oropharyngeal cancer. The locations of primary tumor site were tongue (n = 48, 43.6%), buccal (n = 34, 30.9%), gum (n = 12, 10.9%), retromolar (n = 6, 5.5%), mouth floor (n = 5, 4.6%), lip (n = 4, 3.6%), and hard palate (n = 1, 0.9%), respectively.

### 2.2. Sample Preparation

Aqueous extracts of tissue were resuspended in 650 μL buffer (7.5 mM Na2HPO4, 0.008% TSP, 0.2 mM NaN3, and 92% D2O). The samples were centrifuged at 277 K for 5 min, and 600 μL of the supernatant was transferred into a 4-inch SampleJet nuclear magnetic resonance (NMR) tube (Merck KGaA, Darmstadt, Germany). A plasma sample was constituted by adding 350 mL of the plasma sample and 350 mL of plasma buffer (75 mM Na2HPO4, 0.08% TSP, 2 mM NaN3, and 20% D2O) to an Eppendorf tube. They were centrifuged at 12,000× *g* at 277 K for 5 min, and 600 mL of the supernatant was transferred into a 4-inch SampleJet NMR tube. Eppendorf tubes containing 600 mL of the urine sample were centrifuged at 12,000× *g* at 277 K for 5 min, and 540 mL of the supernatant with 60 mL of urine buffer (1.5 M K2HPO4/KH2PO4, 0.1% TSP, and 2 mM NaN3 in D2O) was transferred into a 4-inch SampleJet NMR tube and mixed well.

### 2.3. NMR Experimental Protocols

The spectrometer contained a Bruker AVANCE III HD (Bruker BioSpin GmbH, Rheinstetten, Germany) console combined with a 14.1-T magnet for 1H 600 MHz. It was equipped with a 5-mm inverse triple resonance CryoProbe (1H/13C/15N) with cold preamplifiers for 1H and 13C and a z-axis gradient with automated tuning and matching. The spectrometer contained a Bruker SampleJet system (Bruker Bipspin AG, Fällanden, Switzerland) set to 5-mm shuttle mode with a cooling rack for keeping samples at 279 K. The data were acquired and processed using Topspin 3.2 (Bruker BioSpin GmbH, Rheinstetten, Germany), and experiments were run under automation using the IconNMR program (Bruker BioSpin GmbH, Rheinstetten, Germany). The experiments were conducted using standard pulse sequences, namely noesygppr1d, cpmgpr1d, and zg30 (Topspin 3.2, Bruker Biospin GmbH, Rheinstetten, Germany) [19].

### 2.4. Data Analysis

For NMR data processing, the line broadening was set to 0.3 Hz, and a zero-filling by a factor of 2 was used to produce 128k Fourier domain points. The H1 of glucose and tetramethylsilane (TMS) was calibrated to 5.23 and 0.0 ppm, respectively. In the score plots of principal component analysis (PCA), the regions around the residual water (5.00–4.20 ppm) were excluded from the analysis. The regions downfield to 10.0 and upfield to 0.3 were also excluded from the analysis. The remaining variables were binned with a bin width of 0.01 ppm using AMIX (Bruker BioSpin GmbH, Rheinstetten, Germany). PCA was conducted on the MetaboAnalyst 3.0 website (www.metabonanlyst.ca) and SIMCA-P 13.0 (Umetrics, Umea, Sweden). The data were then analyzed through PCA and partial least squares discriminant analysis (PLS-DA) using MarkerLynx XS (Waters, Milford, MA, USA) and MetaboAnalyst 3.0 (http://www.metaboanalyst.ca). The variable importance in projection (VIP) values of each variable in the model indicated its contribution to the classification, and higher VIP values represented a greater contribution to the discrimination between groups. A VIP value of >1.2 was considered significant. The results are expressed as mean ± standard deviation for continuous variables and number (percentage) for categorical variables. Where appropriate, data were compared using the Student’s *t*-test, analysis of variance (ANOVA), or the chi-square test. A *p* value of <0.05 was considered significant. The association of metabolites and pathway was informed using the significant metabolites of this study from MetaboAnalyst website and KEGG pathway database (genome.jp/keg). The links of different pathways was built by using metabolites as nodes among pathways.

## 3. Results

### 3.1. Patient Demographics

In total, 110 patients with OSCC were included, with a median age of 52.4 years (range 28–79). We enrolled patients treated from December 2014 to November 2017, according to the American Joint Committee on Cancer (AJCC) Staging System, seventh edition. There were four types of specimens, namely tumor, adjacent normal tissue, plasma, and urine, for patients with each stage of OSCC based on the AJCC Cancer Staging Manual (Table 1). Not all four specimens of each patient were collected. No significant difference in demographics was observed between the staging groups or the four types of specimens. The area, volume, and invasion depth of tumors increased accordingly. The average number of metastatic nodes identified as ENE was 2.9 ± 2.5 (mean ± standard deviation) (Table 2).

### 3.2. Metabolic Alteration in Tumor versus Adjacent Normal Tissue

The PLS-DA plot demonstrated a clear separation of metabolite concentration distribution in tumor against adjacent normal tissue (Figure 1A,B). Regardless of the ENE status, significantly higher levels of acetone, alanine, choline, glutamate, glycine, isoleucine, lactate, leucine, phenylalanine, pyruvate, tyrosine, and valine were found in tumors than in adjacent normal tissue (T-test, *p* < 0.05). However, in patients with stage IV cancer, compared with normal tissue, significantly higher levels of adenosine, aspartate, guanosine, and succinate but significantly lower levels of carnitine, glucose, and sucrose were found in ENE negative tumors (T-test, *p* < 0.05), whereas significantly higher levels of butyrate, glutathione, and O-acetylcholine were observed in ENE positive tumors (T-test, *p* < 0.05; Table 3).

### 3.3. Metabolic Alteration in Tumor with ENE through Tandem Tissue, Plasma and Urine Analysis

A substantial overlap of metabolomics was observed between ENE positive and ENE negative tumors on the PLS-DA plot (Figure 1C). Nonetheless, higher levels of aspartate, butyrate, carnitine, glutamate, glutathione, glycine, glycolate, guanosine, and sucrose but lower levels of alanine, choline, glucose, isoleucine, lactate, leucine, myo-inositol, O-acetylcholine, oxypurinol, phenylalanine, pyruvate, succinate, tyrosine, valine, and xanthine were found in stage IV tumors with an ENE positive status than in tumors with an ENE negative status. The carnitine level, which significantly increased (ANOVA *p* = 0.022) in the tumors with the increase in the tumor stage, demonstrated a significant decrease in adjacent normal tissue (*t*-test *p* = 0.021) (Figure 2B). Decreased levels of alanine and pyruvate were also observed in the tumors with the increase in the tumor stage (Figure 2A,C). In patients with stage IV ENE positive tumors, the plasma levels of creatine, creatine phosphate, glycine, and tyramine significantly decreased compared with the levels in patients with stage IV ENE negative tumors (*t*-test *p* = 0.017, *p* = 0.025, *p* = 0.037, and *p* = 0.034, respectively). Decreased tissue levels of alanine were also observed in plasma and urine samples, although they were not statistically significant.

## 4. Discussion

Metabolomics represent a promising approach for the discovery of novel targets and biomarkers in head and neck squamous cell carcinoma [10]. Previous NMR-based serum metabolomics studies revealed the levels of glutamine, propionate, acetone, and choline accurately differentiated patients with OSCC from healthy controls, with an area under the receiver operating characteristic curve (AUC) of 0.97 [20]. Gupta et al. reported another NMR-based metabolic fingerprint obtained for OSCC, which was significant even at the leukoplakia stage, with the four biomarkers (glutamine, acetone, acetate, and choline) achieving an AUC of 0.96 and an accuracy of 92.4% [21]. In addition, an increase in lactic acid, choline, and glucose and a decrease in proline, valine, isoleucine, aspartic acid, and 2-hydroxybutyrate are associated with the characteristic mechanisms of oral cancer development [16]. In the present study, we demonstrated that the levels of creatine, creatine phosphate, glycine, and tyramine in plasma significantly decreased in stage IV ENE positive OSCC compared with stage IV ENE negative OSCC. Therefore, plasma metabolites can be used as metabolic biomarkers of oral cancer development and can be clinically applied in the early diagnosis and prevention of OSCC. Our experiment result showed that it is possible to distinguish ENE negative and ENE positive tumors in patients with stage IV OSCC using NMR metabolomics biomarkers clinically.

To understand the underlying mechanism behind the alteration of plasma metabolites, our tissue analysis revealed that several metabolic pathways are implicated in the upregulation of metabolites, and the lipid metabolic pathways are involved in the downregulation of metabolites, as summarized in Figure 3 and Figure 4. We found that the carnitine level significantly increased in tumors but significantly decreased in the adjacent normal tissue in advanced stage OSCC, in addition to decreased levels of alanine and pyruvate in tumor tissues. Indeed, carnitine participates in transporting fatty acids across the mitochondrial membrane as a fuel source. It carries activated long-chain fatty acids into the mitochondria matrix for subsequent oxidation and energy production [22]. Our experimental result displayed that the change in carnitine concentration is limited to tumor tissues and adjacent normal tissues. This may suggest that tumors signal the adjacent normal tissue to supply carnitine during proliferation therefore accumulate carnitine in cancer cells.

Our global metabolomics analysis on tumor tissues showed that stage IV tumors with an ENE positive status demonstrated higher levels of aspartate, butyrate, carnitine, glutamate, glutathione, glycine, glycolate, guanosine, and sucrose but lower levels of alanine, choline, glucose, isoleucine, lactate, leucine, myo-inositol, O-acetylcholine, oxypurinol, phenylalanine, pyruvate, succinate, tyrosine, valine, and xanthine than tumors with an ENE negative status. Alanine is closely related to metabolic pathways such as glycolysis, gluconeogenesis, and the tricarboxylic acid (TCA) cycle. Pyruvate is a key intersection between several metabolic pathway networks. Pyruvate can be converted to carbohydrates through the gluconeogenesis pathway, to fatty acids or energy by the acetyl-CoA, and to amino acids alanine and ethanol [23,24]. Therefore, it combines several key metabolic processes. Glutamate is a key compound in cellular metabolism. Glutamine plays a role in lipid synthesis, especially for cancer cells [25]. Guanidinoacetate is a metabolite of glycine, and is transformed into creatine through methylation [26]. Isoleucine is synthesized from pyruvate. After transamination, isoleucine can be converted to succinyl-CoA and introduced into the TCA cycle for oxidation or conversion to oxaloacetate for gluconeogenesis. It can also be converted to acetyl-CoA and can enter the TCA cycle by condensation with oxaloacetate to form citrate. Leucine is also converted to acetyl-CoA. The degradation of valine occurs through oxidation and rearrangement to form succinyl-CoA, which can enter the TCA cycle. Lactate is constantly produced from pyruvate [27]. Our experimental results indicate that the concentrations of alanine and pyruvate in tumor and adjacent normal tissues are continually consumed for tumor development. They may also indicate that the metabolic reaction is required by the tumor and its surrounding. During the proliferation of tumors, concentrations of choline, glutamate, glycine, isoleucine, and leucine metabolites within the tumors do not change; however, they are constantly consumed in the adjacent normal tissue. To this end, further study is warranted to verify if the abovementioned metabolites might be necessary for malignant transformation.

The limitation of our study is the small sample size of tissue, plasma, and urine available in each category, reflecting the difficulty of obtaining human samples with informed consent. Therefore, multiple comparison correction was not performed in our study. Another study limitation was that the status of oral leukoplakia was not recorded, although this is a clinically diagnosed potentially malignant oral disorder with an increased risk of oral cancer development. Although the results of this manuscript are correlative, as was the clinical study aiming at identifying markers that correlate with tumor staging and metastasis rather than functional experiments, our metabolomics studies on OSCC provided metabolite biomarkers that can differentiate ENE negative and ENE positive tumors in patients with stage IV cancer. In the discovery phase, we observed that the metabolomic profile of OSCC was directly correlated with the ENE status. Bonferroni post-hoc correction is a stringent method and could lead to none of the variables being statistically significant in the multivariate ANOVA test in the present study. Larger studies are still required for validating the utility of these biomarkers and translating these data to clinical applications. In our future works, a clinical correlation study of survival will aim to understand what metabolites affect cancer progression and patient prognosis.

## 5. Conclusions

We concluded that metabolomic alterations in tumor tissues correspond to an increase in the tumor stage and are detectable in plasma samples. The metabolomic alterations of OSCC can serve as potential diagnostic markers and predictors of ENE in patients with stage IV OSCC.

## Figures and Tables

**Figure 1 jcm-09-00951-f001:**
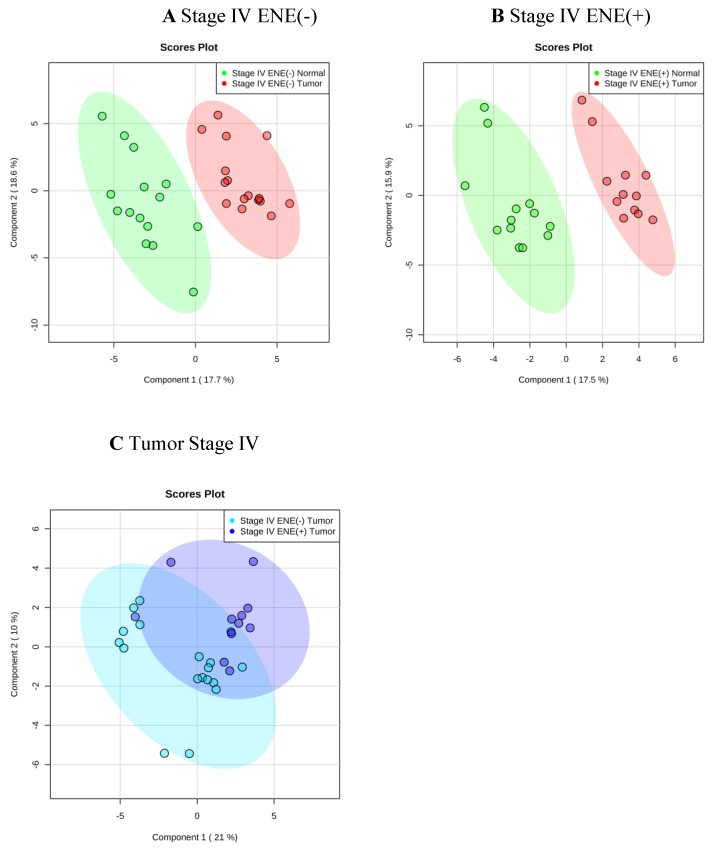
Metabolite-distribution of tumor and adjacent normal tissue at stage IV according to extranodal extension (ENE) status: ENE (-) (**A**) and ENE(+) (**B**). Metabolite-concentration distribution of tumor and adjacent normal tissue at stage IV using the partial least squares discriminate analysis statistical method. Elliptical shape displayed 95% confidence region. (**A**,**B**): tumor/normal tissue = red/green, (**C**): tumor ENE(-)/ENE(+) = cyan/blue.

**Figure 2 jcm-09-00951-f002:**
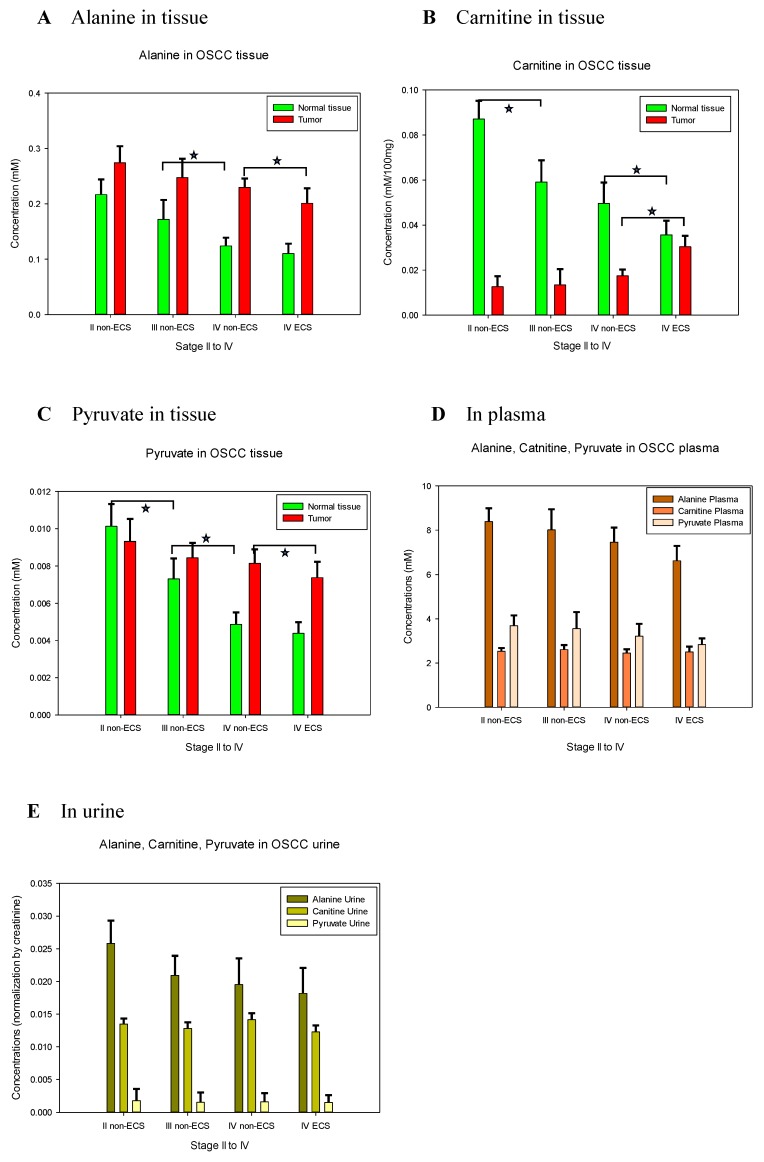
Metabolite-concentrations of alanine, carnitine, and pyruvate in tumor, adjacent normal tissue, plasma and urine from stage II to IV oral squamous cell carcinoma (OSCC) accompany with its extranodal extension (ENE) status: in tissue for alanine (**A**), carnitine (**B**), and pyruvate (**C**); in plasma (**D**) and in urine (**E**). Carnitine level significantly increased in the tumors stage and significantly decreased in adjacent normal tissue. Decreased levels of alanine and pyruvate were observed in the tumor stage and in adjacent normal tissue. Decreased level of alanine was observed in plasma and urine samples. * *p* < 0.05.

**Figure 3 jcm-09-00951-f003:**
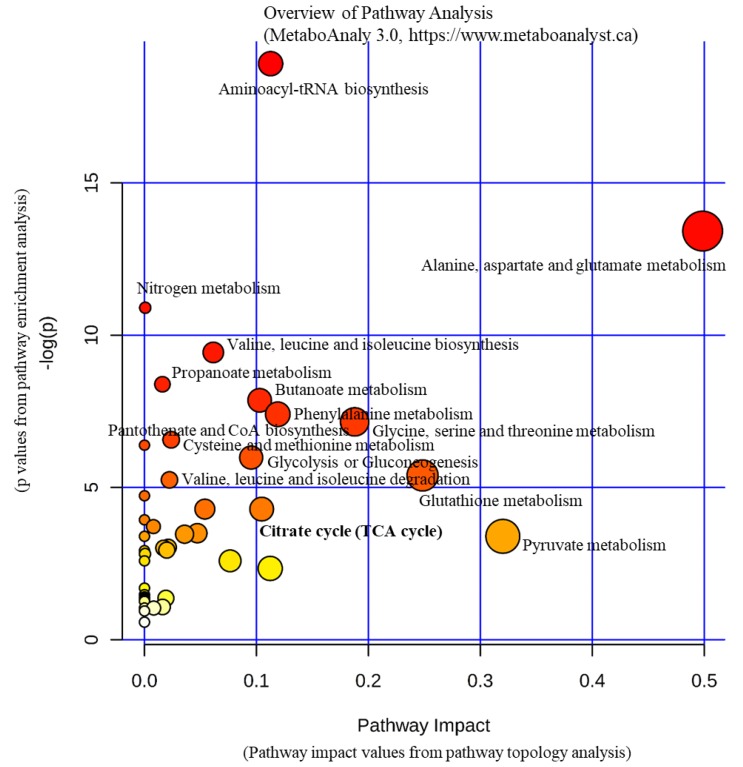
Pathways in oral squamous cell carcinoma tumor. Pathways involving alanine, aspartate and glutamate metabolism, valine, leucine, and isoleucine biosynthesis/degradation, glycolysis/gluconeogenesis, citrate cycle were found in tumors irrespective of extranodal extension status.

**Figure 4 jcm-09-00951-f004:**
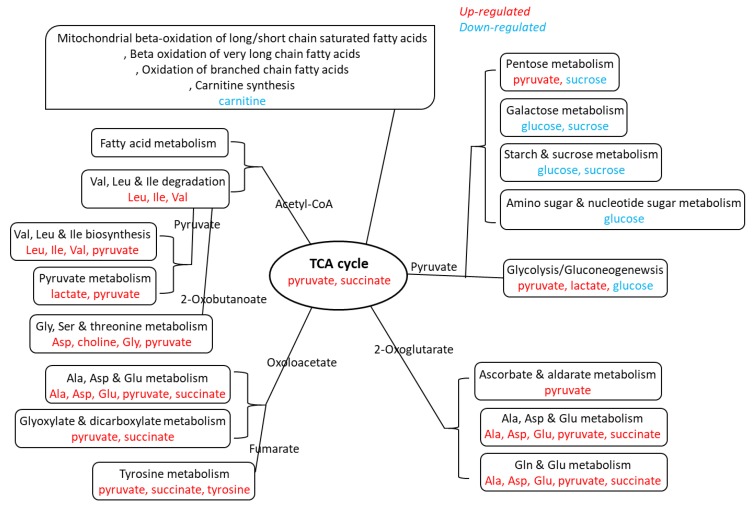
Metabolites in oral squamous cell carcinoma tumor. Upregulated (red) metabolites of pathways (black) involved in tumors irrespective of extranodal extension (ENE) status. Downregulated (blue) metabolites of pathways (black) involved in ENE(-) tumor but not ENE(+) tumor.

**Table 1 jcm-09-00951-t001:** Clinical characteristics of the study (N = 110).

		Tissue		Plasma	Urine
		Normal	Tumor		
Patient Number	110	37	36	44	98
Stage I ENE(-)	18	--	--	7	16
Stage II ENE(-)	25	5	4	15	23
Stage III ENE(-)	15	4	3	4	12
Stage IV ENE(-)	28	15	17	9	26
Stage IV ENE(+)	24	13	12	9	21

Abbreviations: ENE, extranodal extension; ENE(-), ENE negative; ENE(+), ENE positive.

**Table 2 jcm-09-00951-t002:** Area, volume, and lymph nodes with ENE of the tumor.

Tumor	Area (cm)^2^			Volume (cm)^3^			Invasion Depth (cm)			Lymph Nodes (Number) with ENE		
**stage I ENE(-)**	**1.3**	**±**	**0.9**	**0.7**	**±**	**0.9**	**0.45**	±	0.27		--	
stage II ENE(-)	5.0	±	2.9	6.2	±	5.7	1.06	±	0.46		--	
stage III ENE(-)	8.7	±	5.4	11.9	±	10.2	1.06	±	0.54		--	
stage IV ENE(-)	15.7	±	10.1	47.3	±	48.1	2.23	±	1.45		--	
stage IV ENE(+)	17.9	±	16.4	61.3	±	83.7	2.38	±	1.77	2.9	±	2.5

Abbreviations: ENE, extranodal extension; ENE(-), ENE negative; ENE(+), ENE positive.

**Table 3 jcm-09-00951-t003:** Altered metabolites between tumor and adjacent normal tissue at stage IV oral squamous cell carcinoma tumors.

OSCC Tissue	Metabolites		VIP Score	Fold Change (Tumor/Normal)	*p*-Value (T-Test)
Stage IV ENE(-)	Increased	Acetone	1.40	2.08	0.003
		Adenosine	1.10	3.28	0.030
		Alanine	1.19	1.86	0.001
		Aspartate	1.52	1.79	0.001
		Choline	1.87	2.96	4.13 × 10^−7^
		Glutamate	2.02	3.61	1.25 × 10^−6^
		Glycine	1.92	3.78	0.001
		Guanosine	1.70	2.90	1.81 × 10^−4^
		Isoleucine	1.32	1.97	0.001
		Lactate	0.94	1.01	0.032
		Leucine	1.88	2.79	1.81 × 10^−7^
		Phenylalanine	1.57	2.42	7.40 × 10^−5^
		Pyruvate	0.89	1.67	0.035
		Succinate	1.78	9.70	0.006
		Tyrosine	1.57	1.89	8.14 × 10^−5^
		Valine	1.83	2.55	1.23 × 10^−7^
	Decreased	Carnitine	1.50	0.35	2.23 × 10^−8^
		Glucose	1.22	0.67	0.035
		Sucrose	0.88	0.64	0.050
Stage IV ENE(+)	Increased	Acetone	0.97	1.26	0.044
		Alanine	1.35	1.83	0.001
		Butyrate	0.90	1.63	0.046
		Choline	1.83	2.91	1.62 × 10^−4^
		Glutamate	2.10	3.53	5.66 × 10^−7^
		Glutathione	1.09	1.88	0.034
		Glycine	2.85	6.79	3.67 × 10^−6^
		Isoleucine	1.45	2.13	0.001
		Lactate	1.41	1.90	4.04 × 10^−5^
		Leucine	1.52	2.30	1.12 × 10^−4^
		O-Acetylcholine	0.90	1.58	0.032
		Phenylalanine	1.76	2.74	8.81 × 10^−5^
		Pyruvate	0.98	1.68	0.036
		Tyrosine	1.78	2.77	7.57 × 10^−5^
		Valine	1.99	3.02	1.74 × 10^−7^

Abbreviations: OSCC, oral squamous cell carcinoma; ENE, extranodal extension; ENE(-), ENE negative; ENE(+), ENE positive; VIP, variable importance in projection.

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
