# Peer review of "Nuclear Magnetic Resonance Metabolomics Biomarkers for Identifying High Risk Patients with Extranodal Extension in Oral Squamous Cell Carcinoma"

_jcm, 2020, doi:10.3390/jcm9040951_

Round 1
Reviewer 1 Report
the authors satisfied all the comments
Author Response
Thank you very much.
Reviewer 2 Report
The authors replied to most of the comments.
I think that the discussion should state more clearly what the relevance of this work is compared to what already exists in the literature - what would be the clinical advantage of the method proposed?
Author Response
Thank you for your comment. We edited the discussion to state more clearly what the relevance of this work is compared to what already exists in the literature, and also what would be the clinical advantage of the method proposed. Please find attached the version highlighted the changes. Thank you very much indeed.

This manuscript is a resubmission of an earlier submission. The following is a list of the peer review reports and author responses from that submission.
Round 1
Reviewer 1 Report
The paper is of interest with new suggestions according to the last findings and updates in AJCC tumor staging system.
Minor issues should be addressed.
For the used consumables and materials, please specify the name of the seller. For the NMR experimental protocols, please report e reference in order to justify the variables. In paragraph 2.1 is reported 200 patients. In results the analysis were performed in 110 patients. Please elucidate this reduction, exclusion, inclusion criteria? In paragraph 3.1 please specify the edition of AJCC 8th ? The number of sample in table 1 is misleading, it is confusing how many patients were included and for them, which samples were used. When reporting the p-value in results section, please also report which test was used to obtain that result. (For example : Chi-square test, p-value 0,05). Please specify the subsite of OSCC for each patient and report in table. Soft palate and base of the tongue should be removed from the cohort, since they are more related to oropharynx and are at higher risk of HPV infection. English language should be revised, above all in discussion section. You state: four biomarkers (glutamine, acetone, acetate, and choline) were distinguished accurately 183 (AUC, 0.96), and 92.4% were found in oral leukoplakia cases of OSCC. It is not clear. Also specify how many patients reported oral leukoplakia. Please perform Bonferroni post-hoc correction for p-values in ANOVA. Might be of interest performing a multivariate ANOVA, according to the sample studied (tissue, urine or plasma) and ene status or stage. How the pathway has been built? Please specify in methods section.
Reviewer 2 Report
The authors report differences in the metabolites detected in patients affected by OSCCs that depend on the tumor staging. The analysis was performed on tumour tissues, plasma, and urine.
The topic is of clear interest, as abundant evidence supports now a fundamental role for metabolic switched in cancer onset and progression.
The results of the manuscript are completely correlative, as no functional experiment nor explanation is provided. This limitation could be acceptable in a clinical study aiming at identifying markers that correlate with tumor staging and metastatization. The main issue however consists in the lack of recognition of previous works dealing with the same goal, i.e. characterizing the metabolic signature of HNSCC / OSCC and its correlation with tumor staging and metastatization.
E.g.:
https://www.sciencedirect.com/science/article/pii/S221464741630068X?via%3Dihub
Topic Reviewed in:
In the light of these considerations, the study appears completely correlative and superficially discussed.
the language should be improved, and the succession of the topics, particularly in the introduction, more logically organized.